# Effect of Dietary Supplementation with Calcium, Phosphorus and Vitamin D_3_ on Growth Performance, Nutrient Digestibility, and Serum Biochemical Parameters of Growing Blue Foxes

**DOI:** 10.3390/ani12141814

**Published:** 2022-07-15

**Authors:** Jiayu Liu, Zhiheng Du, Ting Li, Yinan Xu, Jing Lv, Xiujuan Bai, Yuan Xu, Guangyu Li

**Affiliations:** 1College of Animal Sciences and Technology, Northeast Agricultural University, Harbin 150030, China; y1719511587@163.com (J.L.); dzhh119@163.com (Z.D.); s200501025@neau.edu.cn (T.L.); 13332407819@163.com (Y.X.); lvjing@neau.edu.cn (J.L.); bxiujuan630306@163.com (X.B.); 2Branch of Animal Husbandry and Veterinary of Heilongjiang Academy of Agricultural Sciences, Qiqihar 161005, China; 3College of Animal Science and Technology, Qingdao Agricultural University, Qingdao 266109, China

**Keywords:** blue fox, calcium, growth performance, phosphorus, vitamin D_3_

## Abstract

**Simple Summary:**

Nutrient requirements have previously been established for minks and foxes, but not specifically for blue foxes. Over the past thirty years, an increase in body size was observed in the farmed blue fox because of the development of modern breeding technology. Hence, it is necessary to study the interaction mechanisms and requirements of calcium (Ca) and vitamin D_3_ (VD_3_) according to the body weights and conditions of farmed blue foxes to guide modern farms. The objective of the present study was to study the influence of dietary Ca and VD_3_ supplemental levels on the nutrient digestibility, growth performance, and serum biochemical indices of growing blue foxes. The results indicated that the Ca and VD_3_ doses showed promising effects on growth performance and nutrient digestibility in growing blue foxes and could reduce fecal nitrogen (N) and phosphorus (P) via improvement in protein and P utilization.

**Abstract:**

Based on the randomized design, a 3 × 3 factorial experiment was designed to examine the effects of dietary calcium (Ca), phosphorus (P), and vitamin D_3_ (VD_3_) supplemental levels with a fixed 1.5/1 ratio of Ca to P on the growth performance, nutrient digestibility, and serum biochemical indices blue foxes’ growth. In total, 135 male blue foxes with the age of 60 days were randomly divided into 9 groups each with 15 blue foxes. The blue foxes belonging to the nine treatment groups were fed Ca supplementation (0%, 0.4%, or 0.8%) and VD_3_ supplementation (1000, 2000, or 4000 IU/kg DM). The base diet contained 0.8% Ca and 327 IU/kg VD_3_. The dosage of VD_3_ in blue foxes showed a significant impact on their growth performance (*p* < 0.05). The Ca dosage had a linear effect on the digestibility of the CP and carbohydrates (CHO) (*p* < 0.05). In conclusion, the results indicated that the Ca and VD_3_ doses showed promising effects on growth performance and nutrient digestibility in growing blue foxes and could reduce fecal N and P via improvement in protein and P utilization.

## 1. Introduction

Calcium (Ca) and phosphorus (P) are the two most abundant minerals in the body, and 99% of calcium and 85% of phosphate present in bone as microcrystalline apatite in the body [1]. Not only the formation of the structural matrix of bone, but also nerve conduction, muscle contraction, and blood coagulation are influenced by Ca and P [2]. In animal diet formulations, Ca and P are important nutrients, especially for bone formation and as enzyme cofactors when proper amounts of them are consumed by livestock [3]. For chickens, biomechanical parameters of bones significantly decrease as the dietary concentrations of Ca and P reduce [4]. In addition, the excessive levels of Ca and P not only affect the absorption and deposition of these minerals in animals, but also the absorption and utilization of other elements. Hall et al. found that pigs fed higher dietary Ca levels (2.7%) died of internal hemorrhage within the initial 28 d of the experiment [5].

There are interactions between Ca and P in the process of absorption and utilization. P supplemented in the diet promotes significantly greater improvement in all the bone variables measured, as well as greater body weight gain and diet and calcium utilization, compared to animals supplemented with calcium only [6]. However, high phosphorus intakes have been suggested to contribute to hypocalcemia (low serum calcium levels) and fractures in children [7]. On the other side, the proportion of Ca/P in the diet affects the absorption of Ca and P. Narrowing the dietary Ca: total P ratio from 2.0:1 to 1.2:1 led to an approximate 16% increase in phytase efficacy for improving performance, digestibility, bone measurements, and serum Ca levels for weanling pigs [8]. Excessive Ca and P levels and an unsuitable Ca/P ratio can affect not only the absorption and deposition of these minerals in animals but also the absorption and utilization of other elements, and excess P can cause environmental pollution [9].

Vitamin D_3_ (VD_3_) plays a positive role in the absorption and utilization of calcium and phosphorus [10,11]. VD_3_ is involved in processes ranging from sustaining Ca homeostasis in bones and plasma to preventing the development of bone diseases such as rickets and osteoporosis [12]. Vitamin D deficiency in patients with ankylosing spondylitis may indirectly lead to osteoporosis by causing an increase in inflammatory activity [13]. For livestock, intaking with the diet is the primary way for an adequate supply of vitamin D. Kim’s (2011) study suggests that high levels of VD_3_ can increase bone growth and mineral deposition in broiler chicks [14]. Whitehead et al. (2004) thought that the requirements of VD_3_ for broilers were much higher than earlier estimates and may be related to higher calcium requirements of modern broiler genotypes, and suggested that the regulations limiting maximum VD_3_ concentrations in broiler starter diets may need to be reviewed [15].

The blue fox, domesticated from arctic fox (*Alopex lagopus*), belongs to Canidae and is a fur animal with great economic value. Research in 1945 showed that the Ca requirement of a growing fox at 7 to 37 weeks of age is between 0.5% and 0.6% in the dry diet [11,16]. According to a study in 1951, a natural food diet that included 0.82 IU of VD3 per gram is adequate for growing foxes [16,17]. Since the 1980s, the nutrient requirements have been established for minks and foxes, but not specifically for blue foxes [17]. Over the past thirty years, the body size of the farmed blue fox has increased from 10 to 15–20 kg at pelting, because of effective genetic selection, ad libitum feeding, and the high amount of fat [18]. Hence, it is necessary to study the interaction mechanisms and requirements of Ca and VD3 according to the body weights and conditions of farmed blue foxes to guide modern farms. The objective of the present study was to evaluate the effects of dietary Ca and VD3 supplemental levels on the growth performance, nutrient digestibility, and serum biochemical indices of growing blue foxes to provide guidance for modern farms.

## 2. Materials and Methods

Our research was conducted in accordance with the guidelines of the Animal Welfare and Ethics Committee of Northeast Agricultural University. All the protocols were approved by Northeast Agricultural University Animal Care and Use Committee (NEAU-EC20160206).

### 2.1. Animal Diets, Management, and Experimental Design

To eliminate the impact of the gender, a total of 135 male blue foxes of the age of 60 days (when weaning stress was terminated) with an initial BW of 1.93 ± 0.27 kg were randomly divided into 9 groups. Each group comprised of 15 blue foxes. The 9 different experimental diets were fed to the blue foxes, i.e., a 3 × 3 factorial experiment with three supplemental levels of Ca (0%, 0.4%, or 0.8%) and three supplemental levels of VD3 (1000, 2000, or 4000 IU/kg DM) was conducted, and the base diet consisted of 0.8% Ca and 327 IU/kg VD3. The nutrient levels, as well as their composition in experimental diets, are listed in Table 1. The ratio of dietary Ca and P was kept constant (1.5/1.0). Experimental animals were individually housed in conventional cages (1 m × 0.8 m × 0.8 m) in standard sheds with open sides and roofs. The study includes a 10 days adaptation period and a 60 days experimental period. In the 10-day adaptation period, animals were allowed to be accustomed to the experimental diets based on dry and powdery components mixed with water. Throughout the study, blue foxes were provided with free access to water and libitum and fed twice per day, namely, at 7:00 and 15:00.

### 2.2. Sample Collection and Preparation

The sample of the basal diet was randomly sampled by the dietary sampler and thoroughly mixed. Three replicates were measured for each sample. The amounts of feed given and that of leftover feed collected daily during the experimental period were weighed. The body weights (BWs) of the animals were measured before feeding in the morning on the 1st, 15th, 30th, 45th, and 60th day. After weighing on day 30 of the trial, eight blue foxes of each group were randomly used to carry out the digestibility experiment. The three days of collection period was used for the feces, feed refusals, and urine. To separate the collection of feces and urine, the animals were kept in metabolic cages [19] and the collection was conducted daily, and was stored at −20 °C till analysis. To prevent the evaporation of ammonia, sulfuric acid (20 mL, 5% solution) was added to the bottles used for urine collection, and the fecal collection trays were sprayed once a day with the same solution of sulfuric acid. At the end of the experiment, all feces from one animal were mixed. Thereafter, 10% feces of the gross weight as well as feed refusals were dried to a constant weight at 65 °C and then ground through a 40-mesh screen. The weight data of the whole feces and feed refusals were collected before and after drying. Urine was preserved with 10% sulfuric acid and 4 drops of toluene, and the urine volume of each animal was recorded. The mixed urine samples were filtered through filter paper, and 10 mL of each filtrate was stored at −20 °C. At the end of the digestibility experiment, blood samples (10 mL) were collected from eight blue foxes from the vein of the hind limb in each group. The collection was made in two separate tubes with a procoagulant substance of 10 μL. The samples were subjected to centrifugation at 2500× *g* at 4 °C for 5 min. The serum was transferred into Eppendorf and stored at −20 °C till analysis.

### 2.3. Analysis of Diets, Feces, and Urine Samples

The contents of nutrients present in the feed and feces were studied using standard methods. The dry matter (DM), crude ash (ASH), crude protein (CP) (CP: Kjeldahl-N × 6.25), fat (EE), Ca, and P contents were determined according to AOAC procedures [20]. Ca contents were determined by the EDTA versanate complexometric titration method. P contents were measured by the molybdovanadate method, and prior to elemental analysis, dry-ashed samples were dissolved in a mixture of HCl and HNO3 (4:1, *v/v*) by heating. The carbohydrate content (CHO) was calculated via subtraction of ASH, CP, and EE contents from the DM content. The CP, EE, and CHO concentrations were used to calculate the metabolizable energy (ME) content of the feed; the reference digestibility coefficients and digestibility were obtained from the NRC1982 nutrient requirements for mink and foxes [17].
ME (MJ/kg) = (4.5 × 0.85 × CP% + 9.5 × 0.90 × EE% + 4 × 0.75 × CHO%) × 4.2/100(1)
N deposition = N intake-fecal N-urinary N(2)
Net protein utilization (NPU) (%) = N deposition/N intake × 100%(3)
Biological value (BV) (%) = N deposition/(N intake-fecal N) × 100%(4)

### 2.4. Analysis of Serum Samples

The serum biochemical and hormone indices including total protein (TP), Ca, P, alkaline phosphatase (ALP), parathyroid hormone (PTH), calcitonin (CT), and 25-OH-D3 were measured with the help of a standard kit (Nanjing Jiancheng Biotechnology Co., Ltd., Jiangsu, China), which the catalogue number were A045-2-2, C004-2-1, C006-1-1, A059-1-1, H207, H153, and H191-1, respectively.

### 2.5. Statistics

The average daily gain (ADG), average daily feed intake (ADFI), and feed conversion ratio (F:G) were calculated for each animal using the following formulas: ADG = (final weight − initial weight)/days(5)
ADFI = sum of daily feed intake/days(6)
F:G = ADFI/ADG(7)

All data are expressed as the mean ± SEM. The data were subjected to ANOVA in a completely randomized design with a 3 × 3 factorial arrangement of treatments using the GLM procedure in the SAS software (SAS Inst. Inc., Cary, NC, USA). All statements of significance are based on a probability of <0.05, and P values between 0.05 and 0.1 were considered a trend. The described model was used for analysis:Yijk = μ + Ai + Bj + (AB)ij + εijk(8)
where Yijk is the measured characteristic, μ shows the overall mean, and Ai and Bj describe the main dietary Ca and VD_3_ effect, respectively; (AB)ij is the effect of the Ca and VD_3_ interaction, and εijk is the random error term. For the significant interactions, the effects of the main factors were avoided while the mean values of the mean were separated by using the PDIFF option of LSMEANS in SAS.

## 3. Results

### 3.1. Growth Performance

The analysis revealed that the dietary VD_3_ level had a significant effect on the final BW, ADG, and F/G (*p* < 0.05, Table 2). The animals fed with mid-level VD_3_ showed maximum final BW, ADG, and the minimum F/G values avoiding any interaction between Ca and VD_3_ levels (*p* > 0.05).

### 3.2. Digestibility

There were no significant interactions between Ca and VD_3_ levels were observed in the digestibility of DM, CP, EE, and CHO (*p* > 0.05, Table 3). The dosage of Ca showed a linear effect on the digestibility of CHO and CP (*p* < 0.05). While the EE digestibility initially increased and subsequently decreased with increasing Ca levels (*p* < 0.05).

### 3.3. N Metabolism

The Ca dose showed a linear increase in NPU and decrease in N intake and fecal N (*p* < 0.05, Table 4). The middle VD_3_ dose was capable of the lowest fecal N and the highest N intake, N deposition, NPU, and BV (*p* < 0.05). There were significant differences in N intake, N deposition, NPU, and BV of protein in the Ca and VD_3_ interaction (*p* < 0.05).

Synergistic behavior was observed between Ca and VD_3_ on N deposition and BV (Figure 1 and Figure 2). The Ca showed little effect on the N deposition and BV but enhanced the N deposition and BV upon the combination with VD_3_ supplementation.

### 3.4. Ca and P Digestibility

The fecal P and Ca and their digestibility showed the interactions between the Ca and VD_3_ levels (*p* < 0.05, Table 5). Fecal P exhibited antagonism between Ca and VD_3_ (Figure 3). The Ca dose linearly increased the fecal P and Ca and decreased the digestibility of Ca and P (*p* < 0.05). The lowest fecal Ca and the highest digestibility of Ca and P were observed with the middle VD_3_ dose (*p* < 0.05).

### 3.5. Serum Biochemical Levels

In the presence of increased dietary Ca, the serum Ca and total protein (TP) increased initially and subsequently decreased, while serum P decreased initially and subsequently increased (*p* < 0.05, Table 6). The middle Ca dosage group showed the highest TP and serum Ca and the lowest serum P. There was no interaction between the Ca and VD_3_ levels for TP, serum Ca, serum P or alkaline phosphatase (ALP) (*p* > 0.05).

### 3.6. Serum Hormone Levels

The dietary level of Ca and VD_3_ have a significant effect on the level of parathyroid hormone (PTH), calcitonin (CT), and 25-OH-D_3_ (*p* < 0.05, Table 7). The Ca dose showed a linear increase in the 25-OH-D_3_ level (*p* < 0.05) while the VD_3_ dose linearly increased PTH and decreased CT (*p* < 0.05). The highest level of PTH and lowest level of CT were observed in the group having mid-level Ca (*p* < 0.05) while the mid-level VD_3_ group showed the lowest 25-OH-D_3_ level (*p* < 0.05). Interactions were observed between the levels of Ca and VD_3_.

## 4. Discussion

### 4.1. Growth Performance

The appropriate levels of vitamin D_3_, calcium, and phosphorus in the diet are very important to the benefit of animal production. More weight was observed for the sows fed 2000 IU of VD_3_ as compared to the sows fed 800 or 9600 IU of VD_3_ [21]. Throughout the finishing of the pigs (35 days postweaning until 135 kg), increasing dietary VD_3_ reduced the F/G (quadratic, *p* = 0.049); and improved the ADG (quadratic, *p* = 0.005). Ca, VD_3_, and *p* are considered crucial for growth and bone integrity, and a shortage of any of these nutrients can result in rickets [22]. Hence, it is quite necessary to provide adequate and balanced vitamin supplementation for suitable growth and an active immune system in mink and other fur-bearing animals [23]. Fat-soluble VD_3_ is particularly sensitive to either deficiency or excess in the diet because the vitamins interact in vivo and in vitro in a dose-dependent manner [24]. In the present study, the initial increase was observed in the final BW and ADG of the blue foxes and decreased subsequently with the increase in VD_3_ level. The best performance was observed in the VD_3_-2000 group indicating that VD_3_ supplementation at 2000 IU/kg was sufficient for blue foxes during their growing period. The results also revealed that either excess or deficient supplementation of VD_3_ will affect the performance of blue foxes.

### 4.2. Digestibility

The increased level of dietary Ca caused the increased digestibility of CP and CHO, showing that Ca has the ability to improve dietary nutrient absorption. Birds fed a 4.3 g Ca/kg diet showed higher N digestibility than birds fed diets with 1.3 g Ca/kg. Therefore, dietary Ca concentration linearly increased the CP digestibility [25]. On the contrary, excessive Ca intake may have adverse effects on the digestion and absorption of nutrients. Sun et al. (2012) found that dietary supplementation of 1.4% and 2.8% calcium significantly decreased the body weight gain and the fat net weight of inguinal fat pad, epididymal fat pad, and perirenal fat pad in high-fat diet mice [26]. This finding is in agreement with the experiment conducted on blue foxes. The increased dietary Ca level initially increased the EE digestibility followed by a subsequent decrease; hence, the excess Ca suppressed the EE absorption. In this study, the ratio of dietary Ca/P was kept constant, while the levels of P were changed by the dietary Ca. Takeuchi (1981) found that the deficiency of dietary P inhibited the β-oxidation of fatty acids [27]. Additionally, excess dietary P promoted the β-oxidation of fatty acids with decreased EE digestibility. It was previously reported that the excess Ca level is susceptible to forming an insoluble saponated substance with fatty acids in the intestines and decreased EE digestibility [28]. In our study, the highest EE digestibility was observed in the mid-level Ca group. The increasing level of dietary Ca showed the initial increase in fat digestibility followed by their subsequent decrease. This is because optimum Ca levels can promote Ca absorption and improve the digestibility of EE, while excessive Ca levels may be associated with increased serum Ca level and Ca deposition in blood vessels and joints. The increase in tissue or organ degeneration and calcification decreased the digestibility of EE. Previous research reported that increasing dietary Ca initially increased the EE digestibility, which was decreased subsequently in growing minks [29]. Therefore, similar results were obtained in the current study. Consequently, the growing foxes have the better capability to digest the CP, EE, and CHO at suitable Ca levels.

### 4.3. N Metabolism

There are certain studies that have investigated the relationship between N metabolism and varying level of dietary VD_3_ and Ca [29,30]. VD_3_ promotes N metabolism, decreases the N excretion in feces, and enhances the BV of protein. These results are consistent with the results of the experiments conducted to examine the effects of different dietary VD_3_ levels on protein utilization and Ca and P digestibility of growing mink in growing minks [30]. This study also revealed that an increase in dietary VD_3_ supplementation level decreased the fecal N content and increased the protein BV and NPU. At the appropriate level of Ca and VD_3_, the growing foxes showed better N deposition, NPU, and protein BV. It may be that an appropriate amount of vitamin D promotes the absorption of phosphorus and improves the utilization rate of nitrogen, while excessive vitamin D increases blood calcium and blood phosphorus, resulting in a decrease in the utilization rate of nitrogen [29]. Furthermore, significant interactions were detected between the intake and deposition of N, NPU, and protein BV with the Ca and VD_3_ levels.

### 4.4. Ca and P Digestibility

A constant ratio of Ca/P was used because both Ca and the Ca/P ratio are involved in bone metabolism [30]. An imbalance in the Ca/P ratio could suppress the absorption of Ca [31,32,33]. In the case of decreased Ca levels, all Ca and P can be absorbed in vivo to meet the nutritional requirement and maintain serum concentration and mineralization of Ca [34]. Therefore, in the presence of maximum Ca and P digestibility, the level of Ca and P should be lowest in feces. If excess Ca is present, then the unnecessary Ca and P are excreted, hence increasing the fecal level of Ca and P, and decreasing the Ca and P digestibility [35]. In the presence of normal Ca levels, the minimum level of fecal Ca and P are excreted with the normalized Ca and P digestibility. Therefore, in our study, three different Ca concentrations resulted in deficient, appropriate, and excess levels. In addition, Vitamin D enhances calcium and phosphate absorption [36]. To perform the function, the VD_3_ must be hydrolyzed into its biologically active form [37]. Over the progression of growth, the VD_3_ is majorly involved in increasing the intestinal absorption of Ca to provide sufficient Ca for bone mineralization [38,39]. Although most VD_3_ is mediated via active Ca transport in the proximal intestine, some VD_3_ has also been found in the ileum and colon [40]. The deficiency of VD_3_ level blocks the Ca absorption, decreases Ca and P digestibility, and increases the fecal Ca and P. Normal levels of VD_3_ facilitate the Ca absorption while the mid-level VD_3_ group showed the highest Ca and P digestibility and the lowest fecal Ca. Both conditions of either deficiency or excess of VD_3_ disrupt Ca and P metabolism in the body and also prevent the growth and development of bone with decreased nutrient absorption [41]. Hence, dietary Ca and VD_3_ levels have significant effects on Ca and P digestibility.

### 4.5. Serum Biochemical Levels

Suitable dietary Ca levels affect TP, serum Ca, and serum P. The observed results are consistent with the broiler experiment showing the increased Ca and ALP in serum upon the influence of increased dietary Ca [42]. In a mouse experiment, restrained serum Ca level was observed in the presence of abundant VD_3_ [43]. ALP is considered an important index for bone metabolism. In a nutritional study on Ca and P in livestock and poultry, the optimum level of dietary Ca and P was determined by ALP. VD_3_ retains the adequate serum concentration of Ca and P, and the intestinal absorption of active Ca^2+^ plays a crucial role in bone mineralization and Ca2+ homeostasis. In conclusion, the excess dietary Ca and VD_3_ restrained serum indices. Hence, the Ca dosage affected the TP, serum Ca, and serum P.

### 4.6. Serum Biochemical Levels

VD_3_ exhibited a vital role in the maintenance of Ca homeostasis. The purpose behind this hormonal control was to regulate the Ca levels in serum and maintain them within a narrow range. For this purpose, coordination with multiple tissues is necessary; these tissues include the intestine, kidney, bone, and parathyroid gland [43]. PTH is capable of performing a variety of key functions related to the Ca balance in bone [44] and kidney [45]. In this study, the PTH was significantly affected by dietary Ca. Similar findings have also been reported previously showing that PTH secretion is dependent on the ionized Ca concentration [46,47]. The metabolism of Ca and VD_3_ are correlated and tightly coordinated through feedback loops established to conserve Ca homeostasis [48]. The VD_3_ dosage increased the Ca absorption even with decreased serum concentration of 25-OH-D_3_ [49]. If there is an excess in the level of serum Ca, the parathyroid follicular cells (i.e., C-cells) reduce the serum level of Ca by the secretion of CT. CT is capable of decreasing the gastrointestinal and renal absorption of Ca and increasing the discharge of Ca in the urine. CT also reported promoting the Ca mineralization of bone. Therefore, serum Ca homeostasis is achieved by the interaction of three hormones (VD_3_, PTH, and CT) [50]. The current study demonstrated that dietary Ca and VD_3_ significantly affected PTH, CT, and 25-OH-D_3_.

## 5. Conclusions

In conclusion, this study determined the optimal dose of Ca and VD3 to promote the growth performance of growing blue foxes by improving the utilization of protein and P while reducing N and P emissions into the environment. Under the conditions of the current study, the recommended dietary Ca was 1.18% (P 0.81%, correspondingly) and the vitamin D3 level was 2327 IU/kg for growing blue foxes.

## Figures and Tables

**Figure 1 animals-12-01814-f001:**
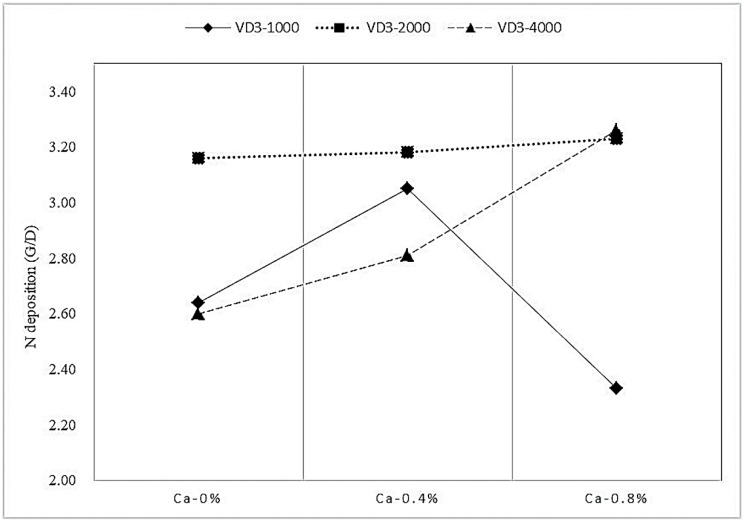
The effect of the interaction between Ca and VD_3_ on N deposition in growing blue foxes. VD_3_ = vitamin D_3_. N = nitrogen.

**Figure 2 animals-12-01814-f002:**
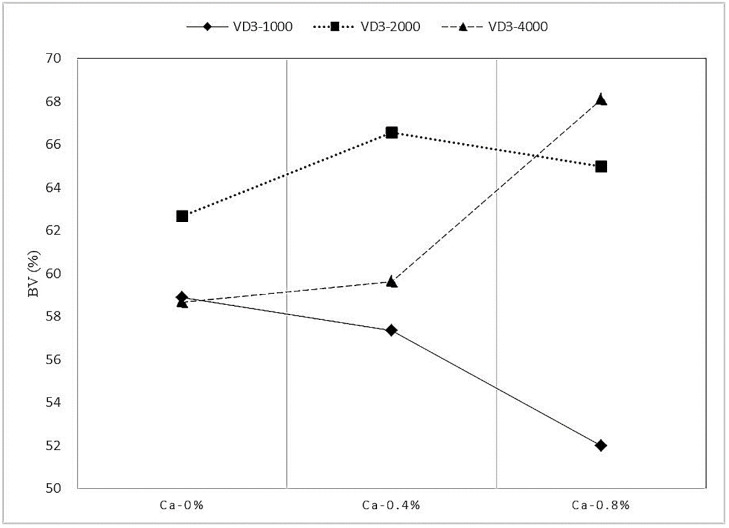
The effect of the interaction between Ca and VD_3_ on the BV of protein in growing blue foxes. VD_3_ = vitamin D_3_. BV = biological value.

**Figure 3 animals-12-01814-f003:**
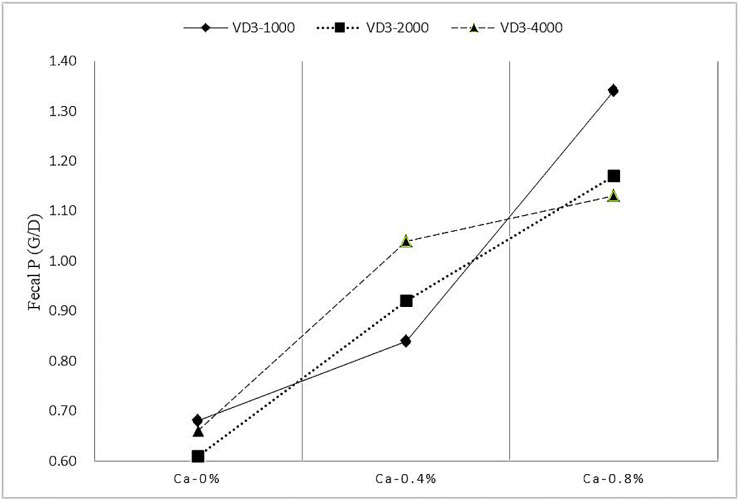
The effect of the interaction between Ca and VD_3_ on the fecal P in growing blue foxes. VD_3_ = vitamin D_3_. BV = biological value.

**Table 1 animals-12-01814-t001:** Composition and nutrient levels of the experimental diet in growing blue foxes (air-dry basis).

Items	Groups								
	I	II	III	IV	V	VI	VII	VIII	IX
Ingredients (%)									
Extruded corn	42.06	42.06	42.06	40.56	40.56	40.56	38.66	38.66	38.66
Soybean meal	20.00	20.00	20.00	20.00	20.00	20.00	20.00	20.00	20.00
DDGS	5.00	5.00	5.00	5.00	5.00	5.00	5.00	5.00	5.00
Corn protein meal	11.24	11.24	11.24	11.34	11.34	11.34	11.64	11.64	11.64
Fish meal	10.00	10.00	10.00	10.00	10.00	10.00	10.00	10.00	10.00
Chicken meal	2.00	2.00	2.00	2.00	2.00	2.00	2.00	2.00	2.00
CaHPO_4_	0.00	0.00	0.00	1.30	1.30	1.30	2.80	2.80	2.80
Limestone	0.70	0.70	0.70	0.80	0.80	0.80	0.90	0.90	0.90
Soybean oil	8.00	8.00	8.00	8.00	8.00	8.00	8.00	8.00	8.00
Premix ^†^	1.00	1.00	1.00	1.00	1.00	1.00	1.00	1.00	1.00
Total	100.00	100.00	100.00	100.00	100.00	100.00	100.00	100.00	100.00
Nutrient levels									
ME ^‡^ (MJ/kg)	13.81	13.81	13.81	13.69	13.69	13.69	13.73	13.73	13.73
CP (%)	28.12	28.12	28.12	28.09	28.09	28.09	28.11	28.11	28.11
EE ^§^(%)	10.82	10.82	10.82	10.81	10.81	10.81	10.79	10.79	10.79
Ca (%)	0.82	0.82	0.82	1.18	1.18	1.18	1.59	1.59	1.59
Total P (%)	0.55	0.55	0.55	0.81	0.81	0.81	1.06	1.06	1.06
Ca/P	1.49	1.49	1.49	1.45	1.45	1.45	1.50	1.50	1.50
VD_3_ ^¶^ (IU/kg)	1 327	2 327	4 327	1 327	2 327	4 327	1 327	2 327	4 327

^†^ Premix provided per kilogram of diet: Fe 80 mg; Zn 60 mg; Mn 15 mg; Cu 10 mg; I 0.5 mg; Se 0.2 mg; Co 0.3 mg; vitamin A 10,000 IU; vitamin E 60 mg; vitamin K_3_ 1.6 mg; vitamin B_1_ 20 mg; vitamin B_2_ 10 mg; vitamin B_6_ 10 mg; vitamin B_12_ 0.1 mg; niacin 40 mg; pantothenic acid 20 mg; folic acid 1 mg; biotin 0.5 mg; vitamin C 120 mg and choline 400 mg. ^‡^ ME was a calculated value, whereas the other nutrient levels were measured values. ^§^ EE = fat. ^¶^ VD_3_ vitamin D_3._

**Table 2 animals-12-01814-t002:** Effects of dietary Ca and VD_3_ supplemental levels on growth performance in growing blue foxes (LS means ± SEM).

Items	Ca Supplementation Level (%)			VD_3_ ^†^ Supplementation Level (IU/kg)			SEM ^‡^	*p*-Value		
	0	0.4	0.8	1000	2000	4000		Ca Level	VD_3_ Level	Ca × VD_3_ Interaction
Initial BW (kg)	1.91	1.92	1.95	1.92	1.94	1.92	0.079	0.810	0.937	0.989
Final BW (kg)	4.11	4.05	4.07	4.00 ^b^	4.17 ^a^	4.07 ^ab^	0.075	0.255	0.049 **	0.806
ADG (g/d)	36.72	35.12	35.39	34.42 ^b^	37.16 ^a^	35.82 ^ab^	9.512	0.128	0.015 **	0.312
ADFI (g/d)	176.37	174.86	173.69	175.20	175.09	174.64	22.497	0.154	0.910	0.381
F/G	4.81	5.08	4.92	5.05 ^a^	4.71 ^b^	4.87 ^ab^	0.193	0.100	0.045 **	0.300

^†^ VD_3_ = vitamin D_3_. ^‡^ SEM pooled standard error of the mean. ^a,b^ In the same row, values with different lowercase superscripts indicate a significant difference (*p* < 0.05), while those with the same or no letter superscripts indicate no significant difference (*p* > 0.05). ** *p* < 0.05.

**Table 3 animals-12-01814-t003:** Effects of dietary Ca and VD_3_ supplemental levels on nutrient digestibility in growing blue foxes (LS means ± SEM).

Items	Ca Supplementation Level (%)			VD_3_ ^†^ Supplementation Level (IU/kg)			SEM ^‡^	*p*-Value		
	0	0.4	0.8	1000	2000	4000		Ca Level	VD_3_ Level	Ca × VD_3_ Interaction
DM digestibility (%)	70.81	71.43	74.27	71.18	73.57	71.31	18.051	0.052	0.129	0.267
CP digestibility (%)	64.52 ^b^	66.76 ^ab^	69.08 ^a^	65.62	68.72	66.07	37.900	0.047 **	0.169	0.132
EE ^§^ digestibility (%)	89.36 ^b^	90.74 ^a^	90.64 ^a^	89.97	90.72	90.17	2.906	0.019 **	0.283	0.291
CHO ^¶^ digestibility (%)	74.40 ^b^	74.85 ^b^	77.45 ^a^	74.96	77.07	74.88	15.969	0.027 **	0.126	0.342

^†^ VD_3_ = vitamin D_3_. ^‡^ SEM pooled standard error of the mean. ^§^ EE = .fat. ^¶^ CHO = carbohydrate. ^a,b^ In the same row, values with different lowercase superscripts indicate a significant difference (*p* < 0.05), while those with the same or no letter superscripts indicate no significant difference (*p* > 0.05). ** *p* < 0.05.

**Table 4 animals-12-01814-t004:** Effects of dietary Ca and VD_3_ supplemental levels on N metabolism in growing blue foxes (LS means ± SEM).

Items	Ca Supplementation Level (%)			VD_3_ ^†^ Supplementation Level (IU/kg)			SEM ^‡^	*p*-Value		
	0	0.4	0.8	1000	2000	4000		Ca Level	VD_3_ Level	Ca × VD_3_ Interaction
N ^§^ intake (g/d)	7.19 ^a^	7.18 ^a^	6.86 ^b^	7.01 ^b^	7.16 ^a^	7.05 ^ab^	0.038	<0.001 **	0.040 **	0.013 **
Urinary N (g/d)	1.81	1.75	1.80	1.89	1.76	1.71	0.156	0.858	0.315	0.674
Fecal N (g/d)	2.52 ^a^	2.35 ^a^	2.09 ^b^	2.41 ^a^	2.12 ^b^	2.40 ^a^	0.162	0.003 **	0.027 **	0.108
N deposition (g/d)	2.81	3.01	2.94	2.67 ^b^	3.19 ^a^	2.89 ^ab^	0.271	0.404	0.004 **	0.023 **
NPU ^¶^ (%)	39.12 ^b^	43.82 ^a^	45.63 ^a^	39.96 ^b^	45.89 ^a^	42.57 ^b^	9.498	<0.001 **	0.007 **	0.033 **
BV * of protein (%)	60.27	62.63	62.54	56.53 ^b^	64.42 ^a^	62.23 ^a^	23.684	0.2901	<0.001 **	0.006 **

^†^ VD_3_ = vitamin D_3_. ^‡^ SEM pooled standard error of the mean. ^§^ N = nitrogen. ^¶^ NPU = net protein utilization. * BV = biological value. ^a,b^ In the same row, values with different lowercase superscripts indicate a significant difference (*p* < 0.05), while those with the same or no letter superscripts indicate no significant difference (*p* > 0.05). ** *p* < 0.05.

**Table 5 animals-12-01814-t005:** Effects of dietary Ca and VD_3_ supplemental levels on Ca and P digestibility in growing blue foxes (LS means ± SEM).

Items	Ca Supplementation Level (%)			VD_3_ ^†^ Supplementation Level (IU/kg)			SEM ^‡^	*p*-Value		
	0	0.4	0.8	1000	2000	4000		Ca Level	VD_3_ Level	Ca × VD_3_ Interaction
Fecal Ca (g/d)	1.00 ^c^	1.72 ^b^	2.26 ^a^	1.71 ^a^	1.50 ^b^	1.78 ^a^	0.093	<0.001 **	0.019 **	0.012 **
Fecal P (g/d)	0.65 ^c^	0.93 ^b^	1.21 ^a^	0.96	0.90	0.94	0.022	<0.001 **	0.445	0.009 **
Ca digestibility (%)	39.59 ^a^	31.08 ^b^	29.19 ^b^	35.60 ^a^	38.86 ^a^	27.30 ^b^	12.264	<0.001 **	<0.001 **	<0.001 **
P digestibility (%)	43.76 ^a^	42.85 ^a^	38.06 ^b^	38.20 ^b^	44.85 ^a^	41.92 ^ab^	0.985	0.010 **	0.008 **	<0.001 **

^†^ VD_3_ = vitamin D_3_. ^‡^ SEM pooled standard error of the mean. ^a–c^ In the same row, values with different lowercase superscripts indicate a significant difference (*p* < 0.05), while those with the same or no letter superscripts indicate no significant difference (*p* > 0.05). ** *p* < 0.05.

**Table 6 animals-12-01814-t006:** Effects of dietary Ca and VD_3_ supplemental levels on serum biochemical indices in growing blue foxes (LS means ± SEM).

Items	Ca Supplementation Level (%)			VD_3_ ^†^ Supplementation Level (IU/kg)			SEM ^‡^	*p*-Value		
	0	0.4	0.8	1000	2000	4000		0.8	1000	Ca × VD_3_ Interaction
TP ^§^ (g/L)	48.83 ^ab^	50.27 ^a^	47.81 ^b^	48.77	49.18	49.01	7.906	0.015 **	0.847	0.085 **
Serum Ca (mmol/L)	2.83 ^b^	3.04 ^a^	2.83 ^b^	2.92	2.96	2.83	0.078	0.018 **	0.204	0.216
Serum P (mmol/L)	2.61 ^ab^	2.50 ^b^	2.66 ^a^	2.51	2.64	2.60	0.043	0.037 **	0.115	0.091 **
ALP ^¶^ (U/L)	176.21	187.37	177.02	180.04	180.06	180.63	4.722	0.585	0.998	0.792

^†^ VD_3_ = vitamin D_3_. ^‡^ SEM pooled standard error of the mean. ^§^ TP = total protein. ^¶^ ALP = alkaline phosphatase. ^a,b^ In the same row, values with different lowercase superscripts indicate a significant difference (*p* < 0.05), while those with the same or no letter superscripts indicate no significant difference (*p* > 0.05). ** *p* < 0.05.

**Table 7 animals-12-01814-t007:** Effects of dietary Ca and VD_3_ supplemental levels on serum hormone in growing blue foxes (LS means ± SEM).

Items	Ca Supplementation Level (%)			VD_3_ ^†^ Supplementation Level (IU/kg)			SEM ^‡^	*p*-Value		
	0	0.4	0.8	1000	2000	4000		Ca Level	VD_3_ Level	Ca × VD_3_ Interaction
PTH ^§^ (pg./mL)	18.83 ^a^	19.05 ^a^	17.39 ^b^	16.73^b^	17.77 ^b^	20.75 ^a^	3.667	0.008 **	<0.001 **	<0.001 **
CT ^¶^ (pg./mL)	15.01 ^a^	12.40 ^b^	12.90 ^b^	14.10^a^	13.59 ^a^	12.66 ^b^	2.076	<0.001 **	0.005 **	<0.001 **
25-OH-D_3_ (ng/mL)	8.65 ^b^	8.87 ^b^	9.98 ^a^	9.64^a^	8.22 ^b^	9.62 ^a^	0.520	<0.001 **	<0.001 **	<0.001 **

^†^ VD_3_ = vitamin D_3_. ^‡^ SEM pooled standard error of the mean. ^§^ PTH = parathyroid hormone; ^¶^ CT = calcitonin. ^a,b^ In the same row, values with different lowercase superscripts indicate a significant difference (*p* < 0.05), while those with the same or no letter superscripts indicate no significant difference (*p* > 0.05). ** *p* < 0.05.

## Data Availability

The data presented in this study are available on request. These data are not publicly available to preserve the data privacy of the commercial farm.

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
