# Peer review of "Effect of Dietary Supplementation with Calcium, Phosphorus and Vitamin D3 on Growth Performance, Nutrient Digestibility, and Serum Biochemical Parameters of Growing Blue Foxes"

_animals, 2022, doi:10.3390/ani12141814_

Round 1
Reviewer 1 Report
Dear authors,
In material and methods it should be described more thoroughly the randomization that is suggested in abstract.
Overall the paper is well edited and presented

Author Response
Response to Reviewer 1 Comments
Point 1: In material and methods it should be described more thoroughly the randomization that is suggested in abstract.
Response 1: Thank you very much for your valuable suggestion. We described the material and methods more thoroughly the randomization (marked in red).
Reviewer 2 Report
Line 20: write in full Ca and VD3
Line 25: write in full N and P
Line 28: write in full P, Ca,
Line 41: add more information regarding foxes farmers and a connected sentences introducing the studies on diet in foxes
Line 42: write in full Ca
Line 43: write in full P
Line 57: add more information regarding blue foxes
Line 66: add the protocol number
Line 69: explain why chose only males and the age of 60 days
Line 69: explain the statistical protocol used to define the sample
Line 84: indicates the site of blood collection
Line 87: add a figure/scheme showing the timeline of the study
Line 97: add catalogue number
Line 99: write in full DM, ASH, CP
Line 102: write in full CHO and ME
Line 198: add a short introductory sentence about the model and the importance in industry
Line 219: add an example also in mammals
Line 239: explain in more details these studies and add references
Line 243: add reference in minks
Line 258: add reference
Line 309: expand conclusions, better explain the best diet chosen
Line 320: add the protocol number
REFERENCES:
Rewrite all references according author’s guidelines
In all the text pay attention to spacing and the justification of the text
Author Response
Response to Reviewer 2 Comments
Point 1: Line 20: write in full Ca and VD3.
Response 1: Thank you very much for your valuable suggestion. We added the full Ca and VD3 (marked in red in Line 18).
Point 2: Line 25: write in full N and P
Response 2: Thank you very much for your valuable suggestion. We added the full N and P (marked in red in Line 23).
Point 3: Line 28: write in full P, Ca,
Response 3: Thank you very much for your valuable suggestion. We added the full P, Ca (marked in red in Line 26).
Point 4: Line 41: add more information regarding foxes farmers and a connected sentences introducing the studies on diet in foxes.
Response 4: Thank you very much for your valuable suggestion. We rewrote the introduction.
Point 5: Line 42: write in full Ca
Response 5: Thank you very much for your valuable suggestion. We added the full Ca (marked in red in Line 39).
Point 6: Line 43: write in full P
Response 6: Thank you very much for your valuable suggestion. We added the full P (marked in red in Line 39).
Point 7: Line 57: add more information regarding blue foxes
Response 7: Thank you very much for your valuable suggestion. We rewrote the introduction.
Point 8: Line 66: add the protocol number
Response 8: Thank you very much for your valuable suggestion. We added the protocol number (NEAU-EC20160206) (marked in red in Line 93).
Point 9: Line 69: explain why chose only males and the age of 60 days.
Response 9: Thank you very much for your valuable suggestion. In order to avoid the influence of gender on the experimental results, researchers prefer to choose single sex as the research object (see references below). In addition, the weaning stress was terminated when blue foxes at the age of 60 days.
References: Zhang T, Zhang H, Wu X, et al. Effects of dry dietary protein on digestibility, nitrogen-balance and growth performance of young male mink[J]. Animal nutrition, 2015, 1(2): 60-64.
Liu Z, Wu X, Zhang T, et al. Influence of dietary copper concentrations on growth performance, serum lipid profiles, antioxidant defenses, and fur quality in growing–furring male blue foxes (Vulpes lagopus)[J]. Journal of Animal Science, 2016, 94(3): 1095-1104.
Point 10: Line 84: indicates the site of blood collection
Response 10: Thank you very much for your valuable suggestion. We added the site of blood collection: vein of hind limb (marked in red in Line 137).
Point 11: Line 87: add a figure/scheme showing the timeline of the study
Response 11: Thank you very much for your valuable suggestion. We added the timeline of the study (marked in red in Line 122).
Point 12: Line 97: add catalogue number
Response 12: Thank you very much for your valuable suggestion. We added the catalogue number in revised paper (marked in red in Line 155-159).
Point 13: Line 99: write in full DM, ASH, CP
Response 13: Thank you very much for your valuable suggestion. We added the full DM, ASH, CP of the study in revised paper (marked in red in Line 143).
Point 14: Line 102: write in full CHO and ME
Response 14: Thank you very much for your valuable suggestion. We added the full DM, ASH, CP of the study in revised paper (marked in red in Line 148, 150).
Point 15: Line 198: add a short introductory sentence about the model and the importance in industry
Response 15: Thank you very much for your valuable suggestion. We added a short introductory sentence about the model and the importance in industry (marked in red in Line 260-261).
Point 16: Line 219: add an example also in mammals
Response 16: Thank you very much for your valuable suggestion. We added an example also in mammals of the study in revised paper (marked in red in Line 282-284).
Point 17: Line 239: explain in more details these studies and add references
Response 17: Thank you very much for your valuable suggestion. We added the references of the study in revised paper (marked in red in Line 455-459).
Point 18: Line 243: add reference in minks
Response 18: Thank you very much for your valuable suggestion. We added the reference in minks in revised paper (reference 30).
Point 19: Line 258: add reference
Response 19: Thank you very much for your valuable suggestion. We added the reference in minks in revised paper (reference 34).
Point 20: Line 309: expand conclusions, better explain the best diet chosen
Response 20: Thank you very much for your valuable suggestion. We rewrote the conclusions (marked in red in Line 371-375).
Point 21: Line 320: add the protocol number
Response 21: Thank you very much for your valuable suggestion. We added the protocol number in revised paper (marked in red in Line 387).
Point 22: Rewrite all references according author’s guidelines
Response 22: Thank you very much for your valuable suggestion. We rewrote all references according author’s guidelines in revised paper (marked in red in Line 397-500).
Point 23: In all the text pay attention to spacing and the justification of the text
Response 23: Thank you very much for your valuable suggestion. We adjusted spacing and the justification of the text in revised paper.
Reviewer 3 Report
1. The introduction is very limited, the physiological interactions of Ca and vit D with the excretion of P and N must be explained.
2. I do not understand why initially 9 diets are indicated with the following levels: calcium (0.82, 1.18, 1.59) and vitamin D (1327, 2327, 4327), and then in results they only indicate the use of 6 diets with 0 Ca? levels. It is assumed that they all included limestone.
3. They do not explain how they distributed the animals in each treatment, I think it was blocking by weight, again it says 9 groups and why only six appear in the results?
4. It is necessary to include details of the digestibility test, consumption, collection of faeces samples, feed rejection, etc.
5. As calcium and phosphorus were measured, it is necessary to include details, the AOAC is not number 9, there is a disorder of the references.
6. Hormones can be measured with the Bradford's method, the reference indicates AOAC?
7. I think it is best to analyze with orthogonal contrasts, including linear and quadratic effects, why was it not used?
8. The conclusion is very general and is not decisive, it does not indicate which levels of Ca and vit D had better results.
Author Response
Response to Reviewer 3 Comments
Point 1: The introduction is very limited, the physiological interactions of Ca and vit D with the excretion of P and N must be explained.
Response 1: Thank you very much for your valuable suggestion. We rewrote the introduction in revised paper (marked in red in Line 39-88).
Point 2: I do not understand why initially 9 diets are indicated with the following levels: calcium (0.82, 1.18, 1.59) and vitamin D (1327, 2327, 4327), and then in results they only indicate the use of 6 diets with 0 Ca? levels. It is assumed that they all included limestone.
Response 2: Thank you very much for your valuable suggestion. The base diet consisted of 0.8% Ca and 327 IU/kg VD3 (marked in red in Line 100-101).
Point 3: They do not explain how they distributed the animals in each treatment, I think it was blocking by weight, again it says 9 groups and why only six appear in the results?
Response 3: Thank you very much for your valuable question. We combined individuals at one level of a factor into a group, and the 6 groups were shown in the results.
Point 4: It is necessary to include details of the digestibility test, consumption, collection of faeces samples, feed rejection, etc?
Response 4: Thank you very much for your valuable suggestion. We described the material and methods more thoroughly (in “2. Materials and Methods” marked in red).
Point 5: As calcium and phosphorus were measured, it is necessary to include details, the AOAC is not number 9, there is a disorder of the references.
Response 5: Thank you very much for your valuable suggestion. The details were added in the “2.3. Analysis of Diets, Feces, and Urine Samples” marked in red, and the number of AOAC was corrected.
Point 6: Hormones can be measured with the Bradford's method, the reference indicates AOAC?
Response 6: Thank you very much for your valuable suggestion. The kit used for serum index determination and its catalogue numbers were described (marked in red in Line 155-159).
Point 7: I think it is best to analyze with orthogonal contrasts, including linear and quadratic effects, why was it not used?
Response 7: Thank you very much for your valuable suggestion. We have tried many statistical methods before and think that the existing statistical methods may be more suitable..
Point 8: The conclusion is very general and is not decisive, it does not indicate which levels of Ca and vit D had better results?
Response 8: Thank you very much for your valuable suggestion. We rewrote the conclusion (marked in red in Line 371-375).
Round 2
Reviewer 3 Report
The authors made the corrections